# Managing obstetric bleeding in Wales: A qualitative evaluation of the OBS Cymru care bundle using Normalisation Process Theory

Tanvi Rai[1]*, Lisa E. Hinton[1], Rosa Mackay[1], Mairead Black[2], Julia Sanders[3], Pauline Slade[4], Amy Elsmore[5,6], Amrit Dhadda[7], William Parry-Smith[5,6], Rachel Collis[7], Stavros Petrou[1], Simon Stanworth[8], Philip Pallman[9], Julia Townson[9], Haddy Fye[10], Ayşe Gür Geden[10,11], Peter Collins[12], Sarah Bell[7]

**1** Nuffield Department of Primary Care Health Sciences, University of Oxford, Oxford, United Kingdom, **2** Aberdeen Centre for Women's Health Research, University of Aberdeen, Aberdeen, United Kingdom, **3** School of Healthcare Sciences, College of Biomedical and Life Sciences, Cardiff University, Cardiff, United Kingdom, **4** Department of Primary Care and Mental Health, University of Liverpool, Liverpool, United Kingdom, **5** The Shrewsbury and Telford Hospitals NHS Trust, Telford, United Kingdom, **6** School of Medicine, Keele University, Keele, United Kingdom, **7** Anaesthetic Department, Cardiff and Vale University Health Board, Cardiff, United Kingdom, **8** Radcliffe Department of Medicine, University of Oxford, Oxford, United Kingdom, **9** Centre for Trials Research, College of Biomedical and Life Sciences, Cardiff University, Cardiff, United Kingdom, **10** Patient and Public Involvement (PPI) co-applicant, **11** UCL Institute of Education, London, United Kingdom, **12** Institute of Infection and Immunity, School of Medicine, Cardiff University, Cardiff, United Kingdom

* tanvi.rai@phc.ox.ac.uk

## Abstract

### Background

Post-partum haemorrhage (PPH) is one of the leading causes of maternal mortality and morbidity worldwide. The Obstetric Bleeding Strategy (OBS) care bundle for PPH management was adopted into Welsh national guidelines in 2019 (as OBS Cymru), and is currently being implemented across 36 sites in the rest of the UK through the OBS UK stepped-wedge cluster randomised controlled trial. We conducted a qualitative evaluation of the OBS care bundle five years after its adoption to inform plans for optimising its implementation across the UK.

### Methods

We conducted ethnographic observations, informal conversations and qualitative interviews with multidisciplinary teams (MDT) in four maternity units in Wales. Data were analysed thematically and using Normalisation Process Theory.

### Results

The OBS Cymru protocol was used daily and MDT members believe it improves the quality and safety of PPH management. The paper proforma supporting OBS Cymru was the 'boundary object' that kept the care bundle in view while clarifying individualised roles across the MDT during a PPH and prompting improved and continuous communication as

**Data availability statement:** The data underlying the results presented in the study cannot be shared as they contain sensitive medical information and we do not have ethical clearance to share these data for projects outside of the study. Contact information for data queries is as follows: Cardiff University's Data Protection Officer, email: inforequest@cardiff.ac.uk or in writing to: Data Protection Officer, Compliance and Risk, University Secretary's Office, Cardiff University, McKenzie House, 30-36 Newport Road Cardiff CF24 0DE.

**Funding:** This project was funded by the National Institute for Health Research (NIHR) (NIHR 152057). Point of care testing machines and consumables were provided free of charge by Haemonetics Corporation and Werfen. CSL Behring provided Fibrinogen concentrate free of charge for the study. These funders had no role in study design, data collection, data analysis, interpretation of data, or writing of the paper. The views expressed are those of the author(s) and not necessarily those of the NIHR.

**Competing interests:** SB, RC and PC developed and led OBS Cymru. SB has received research support from CSL Behring, Haemonetics and Werfen. She has acted as a paid consultant to CSL Behring and received lecture honoraria from Werfen. PC has received research support from CSL Behring, Haemonetics and Werfen. He has acted as a paid consultant to CSL Behring and Werfen. RC has received research support from CSL Behring, Haemonetics, Werfen. All other authors have declared that no competing interests exist. OBS UK receives research support from Werfen, Haemonetics Corps and CSL Behring. OBS Cymru received research support from Werfen. This does not alter our adherence to PLOS ONE policies on sharing data and materials.

bleeding progressed. The standardisation of processes through the care bundle was seen as enabling all staff with an overall knowledge of PPH care, while situating the prominence of their particular roles within a greater whole. Enacting the bundle in practice varied slightly across different settings, according to staffing structures (e.g., in delivery rooms versus theatre births) and caseload, and some residual tensions remained regarding expectations from different staff members and levels of support provided regarding OBS Cymru.

## Conclusions

Despite some small-scale variations, OBS care bundle has become normalised as standard PPH care in Wales. Insights from this evaluation, such as the centrality of the proforma in holding the bundle together, and need for greater clarity in staff role expectations, have informed implementation plans for the OBS UK trial.

## Introduction

Excess bleeding during childbirth is a leading preventable cause of maternal morbidity and mortality worldwide [1], although this burden is overwhelmingly borne by those in low-resource contexts [2]. Post-partum haemorrhage (PPH), defined as a cumulative blood loss greater than 500mL for vaginal birth and 1L for caesarean birth [3], occurs in 50,000 women in the UK every year and around 28% of these require red blood cell transfusion [4–6]. In 2020–22, 18 women died directly due to obstetric haemorrhage in the UK and Ireland [7]. There is also significant morbidity when PPHs progress to massive blood loss (>2.5L), including hysterectomy, women being separated from their baby, and severe psychological consequences [3,8–11]. These outcomes are inequitably distributed, with women of non-white ethnicity and/or from socioeconomically-disadvantaged backgrounds being disproportionally affected [12–14].

The majority of efforts to better manage PPH have been single-intervention approaches however, standardised, integrated approaches that attend to *multiple* aspects of care at different stages of clinical progression are most likely to improve outcomes [15–17]. Previous work, such as the OBS Cymru (the Obstetric Bleeding Strategy for Wales) and the E-MOTIVE trial (conducted in four low and middle-income countries), has demonstrated how effective monitoring systems and an open culture of sustained individual, team-based and institutional learning can produce substantial improvements [16,18]. There is also compelling evidence for the value of regular (and compulsory) refresher training for all members of the multidisciplinary team (MDT), which includes simulation of PPH scenarios [19].

### The OBS Cymru care bundle and the OBS UK trial

Care bundles, which involve a small number (up to five) multi-component and interdependent processes to be enacted by members of a MDT, are known to be effective in changing clinical practice and improve patient care and safety [20,21]. Informed by cumulative research over a few years, a PPH care bundle was developed by multi-professional maternity service stakeholders in Wales and was called the Obstetric Bleeding Strategy (OBS, or locally in Wales, OBS Cymru, Box 1). This care bundle was implemented in all 12 consultant-led obstetric units in Wales between 2017–2018 as a national quality improvement (QI) programme and reported as a service development [16,22]. It led to a 29% reduction in women progressing from moderate to massive PPH (p=0.011) and a decrease in red cells transfused of 7.4% per 1000 per year (p=0.015) [16]. Since 2019, the OBS Cymru care bundle has been incorporated into national Welsh guidance for PPH management, and in Wales, is part of the

PROMPT training that all members of the maternity MDT attend annually (PROMPT, short for Practical Obstetric Multi-professional Training, is a charitable project that provides training for maternity units; helping midwives, obstetricians, anaesthetists and other maternity team members provide care more safely and effectively (https://www.promptmaternity.org)).

> ### Box 1. –Components of the OBS Cymru care bundle:
>
> The OBS Cymru bundle consists of four components, some of which are already incorporated into guidelines used in maternity units across the UK. OBS Cymru necessitates a stage-based approach to be undertaken universally for all births [22]:

a) Risk assessment of every patient to inform planned place of birth and management of the third stage of labour, as per Royal College of Obstetricians and Gynaecologists (RCOG) guidance [5].

b) Cumulative objective measurement of blood loss (volumetric/gravimetric) during every birth to facilitate timely and appropriate escalation of care.

c) Multi-professional working with a senior midwife, obstetrician and anaesthetist attending the mother at 1000 mL blood loss, or earlier for clinical concern, to enable early obstetric interventions to control bleeding, optimise resuscitation and ensure blood sampling for point-of-care (POC) tests of coagulation, haemoglobin and lactate. The two machines used for POC testing are named Rotem or TEG.

d) POC-guided, individualised blood product replacement supported by an algorithm developed from previous research [23,24].

> Alongside a comprehensive set of training packages and drills for local quality improvement teams to implement (and eventually embed) OBS Cymru into clinical practice, a colour-coded paper proforma was also developed to guide the MDT through the four-stage process, prompting risk assessment, cumulative measurement of blood loss, early senior clinical involvement, and POC guided blood product replacement, thus standardising all PPH management and its associated documentation. As we see later, this proforma proved to be a significant "fifth element" of the OBS Cymru intervention.

The success of OBS Cymru for reducing maternal morbidity from PPH and use of blood products [16], led to further funding for the OBS UK trial (www.obsuk.org, ISRCTN 17679951), which is a stepped-wedge cluster randomised controlled trial (RCT) currently ongoing with 36 sites sampled from across the UK (except Wales). This paper reports on a retrospective qualitative evaluation of OBS Cymru undertaken just prior to the commencement of the OBS UK trial. We aimed to explore the Welsh experience of implementing this PPH care bundle five years after its initial adoption, to inform preparations for conducting the RCT and optimising implementation of this same bundle for the rest of the UK.

## Materials and methods

### Design

We conducted a multi-site rapid ethnography [25] which included observations of care, informal conversations and semi-structured qualitative interviews with a range of MDT staff (see S1 Table COREQ checklist). Ethnographic methods are well-suited for the evaluation of complex interventions in healthcare settings [26,27], including in maternity care [28] as it

facilitates a holistic understanding from diverse perspectives of the day-to-day social, cultural, and organizational context.

Our data collection and analysis was guided by Normalization Process Theory (NPT), widely used to understand the implementation, embedding, and integration of new processes and interventions within healthcare settings [29,30]. Examining implementation processes with NPT tools can enable identification of facilitators and barriers to successful implementation, cultural and contextual specificities affecting implementation, as well as local adaptations that facilitate normalisation (Box 2). Ethical approval for this work was granted by the Greater Manchester Central Research Ethics Committee North West – Greater Manchester Central research Ethics Committee (REC 23/NW/0242).

---

### Box 2. –Normalisation Process theory:

NPT focuses on four core constructs to understand the different types of work involved when a new intervention is implemented:

- Coherence: how different stakeholders make sense of the intervention, or the *sense-making* work involved when a new intervention is introduced,

- Cognitive participation: the engagement and involvement of relevant stakeholders, or the *relational* work that is done to create and sustain a community of practice around a new intervention,

- Collective action: the work different stakeholders do to make the intervention work for them, or the *operational* work done to enact the new set of practices around the intervention, and,

- Reflexive monitoring: evaluation and adaptation of the intervention, or the *appraisal* work that stakeholders engage in to assess and potentially reconfigure elements of the intervention according to their contextual realities.

Each construct is constituted of further sub-constructs, some of which we have applied as appropriate to our research needs (for more detail on NPT sub-constructs, see [29,30]).

---

### Data collection

Of the 12 Welsh maternity units, four were identified for evaluation to ensure diversity along the axes of size (annual birthrates), location (rural/urban) and whether it is part of a tertiary and/or teaching hospital or district general hospital (see Table 1). All ethnographic visits and interviews were conducted by TR.

**Site visits.** Between December 2023 and February 2024, TR made single visits to the four sites, for three days each, with total observation time approximately 84 hours. In preparation, the MDT was emailed by the locally-identified principal investigator (PI) explaining the purpose of the visit and a reminder about the study. Study posters were displayed in all areas of the maternity unit explaining that a visiting researcher would be observing care. During visits, TR attended handover meetings where the MDT met at shift changes to go through the clinical case load. With guidance from the Band 7 senior midwife in charge (who had oversight of activities and cases across the whole unit) TR planned where to be present to maximise the chances of observing PPH care. For these higher risk cases, TR requested the attending midwife to introduce her to the patient and take verbal consent for potentially

**Table 1. Characteristics of the four sites.**

| Maternity unit | Annual births/year* | Hospital type | Staff interviewed |
|---|---|---|---|
| Site A | 5331 | Tertiary centre (fetal and maternal medicine) | 3 consultant anaesthetists, 1 band 7 midwife, 1 scrub nurse, 1 consultant obstetrician |
| Site B | 2276 | District general hospital | 2 band 6 midwives, 2 band 7 midwives, 1 consultant anaesthetist, 1 consultant obstetrician, 1 healthcare support worker, 1 anaesthetic registrar |
| Site C | 2481 | District general hospital | 1 consultant obstetrician, 1 consultant anaesthetist, 1 anaesthetic registrar, 1 consultant obstetrician, 4 midwives (across bands 5, 6; two of which also worked as maternity scrub nurses in theatre) |
| Site D | 1959 | District general hospital | 1 consultant anaesthetist, 1 band 7 midwife, 1 locum consultant obstetrician, 3 healthcare support workers |

*Data from 2021,: https://statswales.gov.wales/Catalogue/Health-and-Social-Care/NHS-Primary-and-Community-Activity/Community-Child-Health/birthsliveand-still-by-welshbirthunit-localhealthboard-year)

observing her birth later on. She observed a range of clinical situations (e.g., spontaneous vaginal births, instrumental births, planned and emergency caesarean births) in delivery rooms and operating theatres. In between observing clinical cases, TR situated herself variously in staff rooms, kitchens and handover rooms, carrying out informal conversations with different staff and observing interactions across different members of the MDT.

**Interviews.** In-depth qualitative interviews with members of the MDT including midwives, obstetricians, anaesthetists and healthcare support workers were conducted following a (written or audio-recorded verbal) informed consent process and audio recorded. Interviews were either face-to-face while on site (in staff/ handover/ unoccupied delivery rooms) or online at a later date, when convenient for the participant. Potential interview participants were initially identified by the local principal investigator/point-of-contact, and after spending some time on-site, TR independently recruited additional participants. The topic guide included initial questions about staff role in the unit, followed by questions about PPH care, the OBS care bundle and their experience of how it works in practice.

## Data analysis

With the short timeframe of the fieldwork and the imperative to inform implementation plans for the impending OBS UK trial, we conducted a combined analysis of the four sites to identify cross-cutting themes and patterns in how the OBS Cymru bundle was being implemented across different clinical contexts, looking for variations in sustainability and decay, as well as local adaptations over time. Observational data and informal conversations were captured as fieldnotes [31], in a paper notebook which were further annotated with reflections during the analysis stage, and typed up electronically. All interviews were fully transcribed verbatim and included in the analysis.

A systematic and iterative approach to thematic analysis of the ethnographic notes and interviews was adopted [32]. After multiple readings, TR open-coded the fieldnotes initially, refining them over time and then applying the developing themes to the full set of notes. A table was then

created, organised according to the different themes and populated with data from all four sites. This was then further coded using NPT constructs and sub-constructs and discussed with RM (a qualitative researcher). For the staff interviews, using NVIVO v20, TR and RM independently open-coded a ~10% of transcripts following close reading, and then met to review and resolve overlaps and omissions to create the final coding framework for the rest of the material. As with the field notes, this coding framework included themes from two sets of codes, a thematic set deriving deductively (from the topic guide) and inductively from the data, as well as NPT-based codes. Having these two layers of coding helped to understand the descriptive aspects in delivering PPH care along with more analytical insights to unpack the processes, interactions and communications we were exploring within the study material. The observation notes and staff interviews were analysed in co-constructive ways to provide a deeper and more robust analysis.

**Researcher reflexivity.** TR is an experienced, postdoctoral (non-clinical) ethnographer and external co-applicant leading the process evaluation of the OBS UK trial. She was not involved in the development of the OBS Cymru care bundle, however her access to the sites was arranged by the OBS Cymru/UK team.

## Results

TR conducted interviews with 29 staff across the MDTs. Table 1 shows baseline characteristics of the four sites, and staff interviewed.

Staff across the MDT (midwives, anaesthetists, obstetricians, scrub nurses) reported consistently positive opinions about the OBS Cymru care bundle for managing PPH. Acceptance and engagement was good, and there appeared to be wide agreement that it was a clinically-robust and team-enabling way to manage PPH. During observations, there was evidence across a range of clinical PPH scenarios that the OBS Cymru had become embedded into standard practice: using NPT terminology, the care bundle had become 'normalised' into regular, daily practice. While recently qualified midwives said they only knew the OBS Cymru way of managing a PPH, more experienced staff said they "cannot remember what we did before!". Our results are presented using NPT constructs.

### Coherence: Agreement about value of OBS Cymru, but variation in what it meant

The NPT construct coherence refers to the *sense-making work* that members of a team do, both individually and collectively to understand and then operationalise a new intervention [33]. The fact that OBS Cymru had a clear purpose that aligned with the internalised values of midwifery, obstetric and anaesthetic teams was a major strength of the intervention. They all wanted to improve clinical care and outcomes for patients, reducing morbidity from PPH and managing this childbirth emergency better.

In the early period of OBS Cymru, adopting the 'new' care bundle protocol felt like a burden to some, particularly the need to document (on the associated proforma) every detail of the obstetric care given during a PPH. However, a few years of regular use meant that at the time of this research, staff across MDTs in the four maternity units described the OBS Cymru care bundle as 'actually quite easy to follow' once you got used to it. As one midwife said, it was "almost a bit of an idiot's guide to an emergency, you know?".

Some staff had resisted being challenged on some (mistaken/inaccurate) beliefs they had about their practice:

*I think that as it was introduced, [erm] certainly the obstetricians and theatre staff…..
realised that the... the blood loss that they'd previously been estimating had... was a gross*

*underestimation [...], probably for half of the three year introduction, they were saying, you know, "This... this adding up of blood loss can't be right, you know, [...]I normally have a blood loss of 350mls during my caesareans, and now we're measuring it, it's 700mls." [erm] You know, and it was almost disbelief I think that... [...] even ones that don't bleed excessively were bleeding a lot more than people were thinking,—(S08, consultant anaesthetist)*

Five years of using the OBS care bundle meant that measuring (and not estimating) blood loss was the norm, and staff appreciated how OBS Cymru equipped them to care more attentively, aligning better with patients' particular clinical and time-sensitive needs.

Anaesthetists and obstetricians also noted the significant reduction in wasted blood products at their units. Previously, blood products would be called in based only on a combination of (visually) estimated blood loss and "how well the patient looked" to the clinician or midwife, which they now felt was an imprecise and liberal approach, not without risk. *Differentiation* is an important component of coherence and in these respects, there was keen appreciation for how *different* OBS Cymru was compared to their past PPH management approach. Many staff recalled how before OBS Cymru, although everybody performed their individual tasks to the best of their abilities, the communication and co-ordination of activities among the team was inconsistent. The stage-based 'bundled' nature of OBS Cymru meant that staff now understood the need to communicate constantly to collectively ensure the progressive delivery of different stages of the OBS Cymru protocol.

Staff also appreciated how OBS Cymru integrated well with the PROMPT training [34] they receive annually.

*the PROMPT training days, I... to my mind were introduced at the same time as OBS Cymru, and that was a massive eye opener for the MDT as a whole. I... I feel that those introduced a complete cultural change [erm] and so I'm not sure how long OBS Cymru would have taken to embed if we hadn't had the PROMPT training as well; it probably would have taken a lot longer, I think.— S09, band 7 midwife*

Many staff talked seamlessly about both interventions, using their shared terminology, such as staff becoming too 'task focussed', therefore someone needing to take a 'helicopter view' etc., suggesting compatibility and mutual reinforcement. It seems likely that OBS Cymru was able to ride the wave that PROMPT had initiated, and the shared linguistic terminology/concepts helped staff internalise OBS Cymru better (an essential component of sense-making) and propel its successful adoption.

**But what really is OBS Cymru?.** TR's early ice-breaker conversations with staff, where she explained that she was interested in OBS Cymru, provoked a range of different responses. Some staff would immediately start talking about the 4-page proforma and their practical experience of using it and filling out the various sections, while others would describe the protocolised and close teamworking approach engendered by OBS Cymru.

However, interviews and observations repeatedly reinforced how deeply embedded the OBS Cymru documentation – the physical proforma – was to the successful implementation of OBS care bundle, which went much beyond its obvious practical use. The concept of a 'boundary object' was developed by Star and Griesemer in 1989 to describe objects or entities that allow diverse communities and groups to work *collaboratively* to achieve a shared collective goal [35,36]. These objects may "have different meanings in different social worlds but their structure is common enough to more than one world to make them recognisable, a means of translation" [36; p.396].

In addition to the annual mandatory PROMPT training which, in Wales, includes OBS Cymru in the PPH drills, individual units may also organise local training sessions, where staff can refresh their knowledge of various clinical protocols, including OBS Cymru. However, in the daily case-by-case practice of managing blood loss during childbirth, TR observed how it was the four-page, multi-coloured OBS Cymru proforma divided into different sections, that facilitated the coming together of different members of the MDT to deliver OBS Cymru. It was the boundary object, the *thing* that constantly reminds everybody in the room to think collectively, with detailed cues running through, guiding every stage of clinical progression.

> *It looks like MDT staff 'converge' around the proforma to co-ordinate their individual spe-cialised clinical/care contributions, in order to deliver the collective task of blood loss man-agement, which, of course, occurs alongside other ongoing clinical scenarios. (Observation notes, theatre birth)*

TR saw that the 'doing' and the 'documenting' of OBS Cymru were not separate but deeply entangled, and this physical object held great significance for staff, as we illustrate later.

## Cognitive participation: Identifying roles across the MDT for enacting OBS Cymru

Cognitive Participation is the *relational* work that people do to create and sustain a community of practice around a new intervention [33]. One of its components is *enrolment*, which involves staff considering their individual and collective roles in delivering a new intervention. For some staff, being enrolled into the 'new' OBS Cymru way of managing PPH had not required a dramatic change of practice. Aside from the point-of-care testing (a physical new machine, associated with new procedures and algorithms), they felt they had been following the elements of the care bundle *anyway*, and did not think OBS Cymru constituted something entirely new. This was particularly prominent for obstetricians, for whom the transition to OBS Cymru had not really changed their active role during a PPH event. They were still busy providing hands-on care, but appreciated the shift towards more deliberate ongoing communication with other team members as the clinical situation developed over time.

A couple of anaesthetists expressed approval of the content and format of the proforma as it aligned with their clinical expertise and experience, giving it *legitimacy*, and, as one of them said "anaesthetists like checklists". NPT regards legitimation as essential for staff to feel personally invested in the intervention. Likewise, a midwife noted the value of having a set of prompts that ensured nothing is missed:

> *"…it's great that it prompts you to look up haemoglobin and anticipate if patient will need active management of 3rd stage of labour or cannulation or [another intervention]" (conversation with a band 6 midwife)*

MDT staff believed OBS Cymru had standardised PPH management, and the proforma helped keep in view the care bundle processes *throughout* the birthing process.

> *I think the difference is [er] so the protocol will tell you: 'do this, do this, do this […] like that,' it's just a […] flowchart, […] but the OBS Cymru tell[s] you all the steps and... and also it […] it's designed in a way that you can use it as a documentation at the same time. So that's the step done, or not done, then you tick (S10, consultant obstetrician)*

The NPT sub-construct *activation* speaks to this observation - having collective responsibility for completing the proforma meant that no aspect of care provision could be assumed, as it needed to be manually recorded, line-by-line and initialled by the person filling it out. Furthermore, all members of the MDT could access this information at any time.

> *"So a PPH can be a very chaotic emergency, [erm] and I think the OBS Cymru just gives us a bit of structure so we know who needs to be there and what their job is [erm] and we can easily allocate the jobs to people because we know what's coming next and what we're doing next." (S12, band 6 midwife)*

**Perception of OBS Cymru across roles – Varied responses.** TR asked different MDT staff how OBS Cymru had affected collaborative working across roles, especially in light of different levels of expertise, specialisation and experience. The standardisation of PPH care via OBS Cymru meant that now regardless of their clinical role, each team member would have some top-level knowledge about different PPH situations/stages and what needs to happen according to the protocol. While this may be empowering to some, we were curious about whether it also created new tensions across roles, and in more hierarchical teams, it may have the potential to alter or challenge some team relationships. An anaesthetist responded that the fundamental effect of the OBS Cymru was not so much about "flattening the hierarchy" but more in terms of "getting everyone on board" with delivering the best of their particular expertise in explicitly collaborative ways. So while they deliver their expertise-specific tasks, they are also aware of how they link with other tasks to form the whole bundle of care. Returning to the boundary object character of the OBS Cymru proforma, this 'tacking back-and-forth' between more general and more specialised forms of collective working by interdisciplinary team members is precisely what such an object enables [35].

There were however, some incongruent perspectives. A couple of midwives told TR how OBS Cymru was 'empowering for everyone', citing how it had granted healthcare support workers (HSW) new authority to raise concerns. During a PPH event, HSWs' practical roles include assisting others by measuring blood loss, or being the nominated scribe or runner. These midwives explained that previously, HSWs worried about not understanding clinical and pharmacological terminology and how to spell/pronounce terms and therefore would hesitate to ask questions or request clarifications. However, having the clinical prompts on the OBS Cymru proforma, meant they would just call them out and simply tick and sign the appropriate box to indicate completion.

> *She mentioned how OBS Cymru gave the whole team more confidence to be able to raise concerns, for example previously, if a doctor shouted out 'giving tranexamic acid', the HSWs used to worry about not knowing how to spell it or what it was for etc, and would hesitate to ask again […], whereas the proforma gives them confidence that even if they don't understand what it all means they can fill in the sheet and voice their concerns based on the sheet if something hasn't happened… (Notes from informal discussion with two band 7 midwives)*

Interviewing HSWs themselves presented a contrasting view; the midwives' perceived newfound "confidence" was not evident among them and they felt stressed about the misalignment between what was expected of them, and the level of training and support they had received.

> *"Maybe a bit of training and explaining the importance […] of why we're monitoring the blood loss […] We know to get to 1,000 [mL blood loss], it's like in the red area, but we don't*

*know the severity of this and what's happening 'cause we're not trained enough to know about the [...] boxes that you tick and then it goes to the next section, what's happening, and what to do next; we're really don't know what it is [...] It is pressure, yeah. Because if you miss something, it is on you."* (S27, S28, S29; group interview with 3 HSWs)

## Collective action: Things working in practice

This next NPT construct explores the *operational* work that people do to enact a new set of practices, about how the work actually gets done. As staff had described, TR's observations of PPH care during childbirth confirmed that in order for the OBS Cymru protocol to work smoothly, everybody needed to communicate clearly, calmly and in a time-sensitive way with each other all the way through, e.g., by discussing each patient's PPH risk during ward rounds, handover and 'huddle' meetings and during the completion of the WHO checklist (in theatre births), anticipating a worsening situation that is likely to need intervention and escalation to senior clinicians, verbalising every time new drugs or blood products were administered, calling out contemporaneous blood loss measurements, and when the different blood tests, cannulation and surgical procedures were undertaken. These continuous vocalised updates had direct implications for keeping everybody aware of what OBS Cymru 'stage' the PPH situation was at, and anticipating and preparing for whatever was needed next.

However, appropriate resourcing was essential to enable *contextual integration* of the care bundle across the different birthing environments. The measuring of blood loss occurred differently between births in delivery rooms and operating theatres. In the former, the attending midwife may ask the junior midwife or HSW supporting her (if present) to start weighing the blood loss, especially if there was a gush or a continuing trickle of blood after childbirth. During one observation however, there was only one midwife in the room attending to the mother and baby. She used visual estimation to escalate the situation when she thought the blood loss looked serious (a gush after placenta delivered) and pressed the emergency buzzer. There are no guidelines in Wales or the UK as a whole that stipulate two midwives (or a midwife and a HSW) to be at every birth, but this is usual practice at home births and water-births. In obstetric units, a single midwife may be the only health professional present during birth to care for the mother and baby, assess blood loss and document all events. Having just one person at a birth clearly poses a challenge to following OBS Cymru as intended.

Escalation in the form of calling in senior clinicians was more obvious in delivery room births where care was primarily midwife-led. Operating theatre births were usually for assisted vaginal and caesarean births with multiple clinicians present, their seniority being dependent on the case. Pertinently, escalation events depended on the unit's culture regarding the level of comfort, and acceptability for calling in on-call consultants. While the proforma enabled good communication between the midwives and the clinicians *during* a PPH event, general camaraderie between them appeared to be stronger in units where the 'hanging out' spaces, like kitchens and handover rooms, were actively shared by staff from across professional boundaries, and this is a likely strong influence on ease of escalation.

During births in operating theatres, the responsibility for measuring blood loss lay with theatre staff. Whether they were maternity-specific scrub nurses or 'main' theatre nurses varied across units and had an impact on OBS Cymru implementation. When their roles were not maternity-specific they were less familiar with OBS Cymru, as described in the following communication with midwives:

*"I think the theatre team are not as quite on board, because they're nurses and not midwives, [...]...they're just not as invested in it as the midwifery team, [erm] and this was the place it*

*was hard to sell it to because it's like just a whole separate team […]… they don't fill it in… it's a tricky one because you… you need that kind of buy-in because it isn't just weighing it, it's the what the actions are from that, you know?[…] (S07, band 6 quality improvement midwife)*

*"…their expertise, they're scrub nurses, and more so then will always like to weigh five swabs together". In those situations, the attending MW might call the band 7 to escalate ("using the hierarchy") – so. Band 7 will now come into theatre and tell the scrub nurses "to measure what we've got" (Observation notes following informal conversation with midwives)*

TR observed the use of the point-of-care machines to test for coagulation just a small number of times (all during theatre births, and not at all sites) when blood loss was greater than 1L. Each time, the process was initiated by the anaesthetist present, and either the anaesthetic assistant or the band 7 midwife would take the sample to process on the machine (stored in another room, usually sat alongside a blood gas machine).

**Filling out the proforma.** Developers of OBS Cymru had consciously avoided being prescriptive about specific roles, recognising that different units may be structured and staffed differently, and should have autonomy to allocate roles accordingly. As such, when asked, staff hesitated to name any particular team member/role whose job it was to fill out the OBS Cymru proforma, despite all agreeing that it would get done. A few said it had to be someone who has not got another job and has their hands free, and this could change over the course of caring for the same patient. Some midwives felt that anaesthetists should take more responsibility for it:

*"…anaesthetists sometimes are a little bit more reluctant to fill in a form, so then it usually gets [erm] kind of on the midwife who's… yeah, tends to get lumped with all other stuff" (S11, band 7 midwife)*

Senior (band 7) midwives felt burdened with the responsibility of overseeing OBS Cymru, especially when they may only be called in when the situation escalates. In delivery rooms, the attending midwife would likely have a HSW or junior midwife assisting them who could fill out the form some of the time. Meanwhile in theatre, the attending midwife would take charge, but once the baby is born, they would need somebody else to take over. Midwives felt more in control in delivery rooms, while in theatre they felt subordinate to clinicians, and frustrated when clinicians forgot to call out what drugs were given or procedures undertaken. In either setting, TR did not always observe the proforma being filled out contemporaneously, suggesting that sometimes it might be filled in later. Midwives from the maternity-led units (MLUs) said they would only start measuring blood loss if it 'looked like a lot' (but admitted this was difficult during water births), and would transfer the patient to the delivery suite when blood loss was over 500mL.

**Proforma enabling co-operation, but occasionally also surfacing tensions.** The NPT sub-construct *relational integration* is 'the knowledge work that people do to build accountability and maintain confidence in a set of practices and in each other as they use them' [33]. The OBS Cymru proforma facilitated this process most of the time. By being the boundary object that everybody coalesced around, either literally or figuratively, it promoted clarity and structure in decision-making. This was not just because it was evidence-based and mandated by guidelines, but also because it was *external* to individual clinical identities. It did not challenge or compromise clinical expertise, and instead offered a standardised, protocolised process and decision aid as a basis for clinical decision-making.

*The OBS proforma is useful tool to hold up if anyone in the room feels something needs to happen and it isn't happening yet, or if there are strong differences in clinical opinion (informal conversation with anaesthetist)*

However, in some situations this protocolised nature of the OBS Cymru bundle had led to frustration across differently specialised staff. These experiences speak to the NPT sub-construct *skill set workability* which explores how the work is distributed among different team members to deliver the intervention. For example, sometimes an obstetrician may feel fully in control of a clinical situation, without the need to escalate, whereas a midwife may feel more bound to what the proforma recommends for that stage of blood loss, as recalled by this anaesthetic assistant:

*He tells me some midwives in theatre "get stressed every time the blood loss crosses a threshold" but the clinicians want to use their clinical judgement to decide whether bloods need to be ordered because sometimes it might cross threshold but then stop bleeding. He then recalls one time when midwife ordered blood products despite the obstetrician's resistance, and they were wasted. (Informal conversation with anasthetic assistant)*

Thus, the convergence around the boundary object was not always to reinforce OBS Cymru's unifying purpose, as in this case, where it caused a communication rupture. It is not entirely unexpected that the overall increased communication that OBS Cymru engenders may sometimes surface clinical disagreement; the overall positivity around OBS Cymru suggested these occasional frictions were not considered very significant by staff.

## Reflexive monitoring: Impact of OBS Cymru on PPH care and team working

This final NPT construct relates to the *appraisal* work that the team does; having used a new initiative for some time, they reflect on the ways in which the new intervention or practices have affected them, individually and as a team, including how they work together now in light of the intervention. Since this work was done retrospectively, much of how people talked about OBS Cymru was in retrospective ways. As with any new intervention being introduced into a multidisciplinary, high-intensity setting, changing practice had not been straightforward: one of the quality improvement midwives remembered feeling demoralised at times, but ultimately glad she persevered:

*"It was really challenging because not everybody wants to change their practice; […] And because when OBS Cymru was brought in, […] it was designed by anaesthetists but very much sort of put at the feet of midwives to kind of roll it out, so I had to be really proactive. And it was a real challenge to be honest, […]–– But […] it got there in the end –... I'm so glad I was part of it 'cause it's fantastic. "(S07, band 6 quality improvement midwife)*

Sustaining OBS Cymru such that it remains standard practice has required continuing training sessions, particularly given high staff turnover rates, but this commitment to ongoing training varied across units, and was driven almost entirely by enthusiastic individuals.

*"we had to keep [erm] doing more training sessions. So every year, since we've introduced OBS Cymru, I've done like simulation training. We've got... got a clinical school with a simulator and I try to do a multidisciplinary team training to have everybody there" (S25, band 7 midwife)*

The need for regular training was clearly essential for new staff, and for those transferring from units outside Wales (where OBS Cymru is not followed). For instance, two locum obstetric consultants who had been working in Wales for a few months could not recall attending any OBS Cymru training.

**Reflecting on OBS Cymru's impact.** Variation existed in aspects of the care bundle most valued across different staff members. Midwives felt that measuring blood loss contemporaneously (instead of estimating it, or measuring it at the end) had been the single biggest change that had significantly improved PPH care. Meanwhile, anaesthetists felt the introduction of point-of-care tests, allowing rapid access to the patient's blood coagulation profile (which enabled clarity and efficiency in blood product administration), was the 'game changer'. As one anaesthetic consultant said: *"I can think of at least four women who would have been dead if it wasn't for the Rotem"*.

Relevant to NPT sub-construct *systemisation*, which is about collecting information to evaluate the usefulness/effectiveness of an intervention, some staff recounted gathering PPH cases together (both well and poorly managed) to talk through them, with their own team or with practitioners from other units. This practice often formed part of regular risk and governance audits, but was useful for appraising how OBS Cymru was being used and what differences it made. In some cases, particular scenarios would also be fed into subsequent PROMPT trainings to consolidate learning with the wider MDT.

> *"…in the first few years when […] the study was... was running, I gathered all our cases and we did just some audit presentation of cases…feed back, so we could learn from each: 'are we getting this right, are we not getting this right?'" (S25, band 7 midwife)*

There was also discussion about the wider unanticipated (positive) outcomes of OBS Cymru. For example, an anaesthetist remarked how OBS Cymru training and proforma use had made the entire MDT a lot more knowledgeable about PPHs. Staff also mentioned positive 'spill-over' effects of OBS Cymru, where it had improved teamworking not just during PPHs but also during other health emergencies.

**Reconfiguration and 'tweaking'.** Some staff mentioned how the OBS Cymru blood loss thresholds for escalation need to be adjusted according to the patients booking weight. OBS Cymru does encourage clinicians to adjust the blood loss thresholds if the booking weight of patients is <55 Kg or they have low BMI. As such, a couple of sites had developed localised protocols, with posters on the walls promoting a reconfigured OBS Cymru protocol such that escalation would occur at different blood loss levels for patients, calculated using their booking weight. However, TR witnessed an elective caesarean section of a patient with high BMI where this personalisation of escalation thresholds proved unsatisfactory. Before the surgery, staff had calculated (and noted on the theatre white board) the blood loss escalation volumes according to their localised formula (these cut-offs were higher than OBS Cymru levels). However, on seeing the patient bleeding continuously, clinicians initiated activation of the massive haemorrhage protocol significantly earlier than their stated thresholds stipulated. We can see how this introduction of uncertainty is unhelpful and distracting, especially during an obstetric emergency like PPH, when ultimately, it is the expertise of clinicians in the room that guides decision-making. It is also important to note that blood volume does not increase proportionately with BMI [37].

## Discussion

While tensions still surface around some details (e.g., who fills out the proforma, cut-offs for escalation), the OBS care bundle has embedded as standard PPH care in Wales and is

valued highly by staff. NPT proposes that complex interventions become routinely embedded in their organisational and professional contexts when people do some work, individually and collectively, to implement them [29]. This requires action, which goes beyond having the appropriate attitude or intentions, and occurs through the four generative mechanisms according to which we have presented our results. Sustaining an intervention such as OBS Cymru such that it is integrated into standard care has required repeated cycles of sense-making, commitment and work from MDT staff and regular appraisals. Despite staff turnover and variation in annual birthrates (and therefore PPHs) across sites, OBS Cymru remains in near-daily use, without becoming fragmented into its component parts. We found that the OBS Cymru proforma was the boundary object helping to operationalise the care bundle, such that the documenting of PPH care via this proforma symbolised not just a record of protocol compliance but the means to hold the OBS care bundle, and the team who deliver it, together.

Similar to other PPH care bundles showing improvements [17,38], OBS Cymru's requirement for full MDT simulation training (incorporated within PROMPT), regular appraisal processes and verbalisation of activities while working through the proforma during PPHs helped reduce the effect of human factors during PPH emergencies [39] as well in dismantling some of the clinical tribalism [40] often found in these settings. Our findings have informed implementation plans for the trial, such as explicitly recommending inclusion of HSWs in OBS UK training and changing wording on the proforma to enable greater flexibility. The crucial role of the proforma in giving meaning to the care bundle was the most significant finding which has been integrated into the implementation training.

Incorporating ethnographic observations and informal conversations with staff alongside formal interviews was invaluable for teasing out what people say from what they actually do [41]. This was particularly important as TR's introduction to sites as someone *interested to hear about OBS Cymru* may have biased who had initially come forward (or were identified) as potential interviewees. This work was a purely qualitative exercise without quantitative measures of fidelity etc., where our aim was to gather practical insights to inform the national trial rollout. We also collected ten patient interviews, but have not included them here as they tended to recall their entire birthing experience (including antenatal and postnatal care), with limited details pertaining to their PPH care specifically.

It remains to be seen whether the promise of the OBS care bundle as seen in Wales can be fulfilled across the different UK contexts and to what extent, or whether there is some Welsh uniqueness (in practical aspects, or for less tangible cultural reasons) that has buoyed the continued success of OBS Cymru. For example, having paper-only notes clearly allowed the OBS Cymru proforma to slot smoothly into existing patient monitoring systems, but most UK sites outside Wales are partially or entirely digital. Patient demographics across Wales tends to be less diverse (particularly regarding ethnicity), and the midwifery staff are mostly Welsh and local. For the rest of the UK, the picture is quite different, particularly in high-density metropolitan settings, where there is significant ethnic and socioeconomic diversity [42] as well as more transient populations.

## Conclusion

The OBS Cymru care bundle continues to form standard PPH care in Wales. National reviews within the UK highlight the need to improve and standardise maternity care across the country, with recent reports finding wide variations in practice and deficiencies in management in 80% of massive PPH cases (19). At the time of writing, the OBS care bundle is being introduced sequentially across 36 sites in the UK. The heterogeneity of included trial sites [42] will ensure a comprehensive effort to recreate the OBS Cymru experience across the diversity

of the UK population. Our future publications will report findings from the OBS UK trial and experiences of implementing the OBS care bundle across the different sites.

## Supporting information

**S1 Table. Completed COREQ checklist.**
(PDF)

## Acknowledgements

We thank all the maternity-based clinicians at the study sites for supporting this work.

## Author contributions

**Conceptualization:** Tanvi Rai, Julia Sanders, William Parry-Smith, Rachel Collis, Julia Townson, Peter Collins, Sarah Bell.

**Data curation:** Tanvi Rai, Rosa Mackay.

**Formal analysis:** Tanvi Rai, Rosa Mackay.

**Funding acquisition:** Tanvi Rai, Mairead Black, Julia Sanders, Pauline Slade, William Parry-Smith, Rachel Collis, Stavros Petrou, Simon Stanworth, Philip Pallman, Julia Townson, Ayşe Gür Geden, Haddy Fye, Peter Collins, Sarah Bell.

**Investigation:** Tanvi Rai.

**Methodology:** Tanvi Rai, Julia Sanders, William Parry-Smith, Rachel Collis, Julia Townson, Peter Collins, Sarah Bell.

**Project administration:** Tanvi Rai, Julia Townson, Sarah Bell.

**Resources:** Tanvi Rai, Julia Sanders, Rachel Collis, Julia Townson, Peter Collins, Sarah Bell.

**Supervision:** Tanvi Rai.

**Writing – original draft:** Tanvi Rai, Rosa Mackay.

**Writing – review & editing:** Tanvi Rai, Lisa E Hinton, Rosa Mackay, Mairead Black, Julia Sanders, Pauline Slade, Amy Elsmore, Amrit Dhadda, William Parry-Smith, Rachel Collis, Stavros Petrou, Simon Stanworth, Philip Pallman, Julia Townson, Ayşe Gür Geden, Haddy Fye, Peter Collins, Sarah Bell.

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
