## [Decision Letter · Decision Letter 0]

25 Feb 2025

Managing obstetric bleeding in Wales: a qualitative evaluation of the OBS Cymru care bundle using Normalisation Process Theory

PONE-D-24-59015

Dear Dr. Rai,

We’re pleased to inform you that your manuscript has been judged scientifically suitable for publication and will be formally accepted for publication once it meets all outstanding technical requirements.

Kind regards,

Alfredo Luis Fort, M.D., M.Sc., Ph.D.

Academic Editor

PLOS ONE

1. Thank you for stating the following in the Competing Interests section:

[I have read the journal's policy and the authors of this manuscript have the following competing interests:

SB, RC and PC developed and led OBS Cymru. PC has received research support from CSL Behring, Haemonectics and Werfen. He has acted as a paid consultant to CSL Behring and Werfen. All other authors have declared that no competing interests exist.].

Please confirm that this does not alter your adherence to all PLOS ONE policies on sharing data and materials, by including the following statement: ""This does not alter our adherence to PLOS ONE policies on sharing data and materials.” (as detailed online in our guide for authors http://journals.plos.org/plosone/s/competing-interests ). If there are restrictions on sharing of data and/or materials, please state these. Please note that we cannot proceed with consideration of your article until this information has been declared.

Please respond by return email with your amended Competing Interests Statement and we will change the online submission form on your behalf.

Additional Editor Comments (optional):

Given the importance of your study and results, we have decided to go ahead and publish it, obviously after making some very small corrections for better reading and understanding by prospective readers (see attached file for a few suggestions for editing, which I also listed below). Thank you.

Page 7, lines 123-124: where it says "...widely used understand...", there needs to be a "to" added between "used" and "understand".Page 7, Box 2: where it says "...different kinds of work involved...", replace "kinds" with "types" (a more formal use of English word).Page 23, line 462: where the note in italics says "...ordered blood products and were wasted...", delete the extra space there.Page 23, line 470: where it says "...work that the team does, having used a new initiative..., change the comma with a semi-colon (;), since it's a long sentence and a new expression different from the previous one.Page 26, lines 538-539: where it says "...to implement them (29) This requires action...", insert a period after the quote, since it's the end of the sentence.

Reviewers' comments:

Reviewer's Responses to Questions

**Comments to the Author**

1. Is the manuscript technically sound, and do the data support the conclusions?

Reviewer #1: Yes

2. Has the statistical analysis been performed appropriately and rigorously? 

Reviewer #1: N/A

3. Have the authors made all data underlying the findings in their manuscript fully available?

Reviewer #1: Yes

4. Is the manuscript presented in an intelligible fashion and written in standard English?

Reviewer #1: Yes

5. Review Comments to the Author

Reviewer #1: The manuscript is succinctly written, especially the methods section, it was done and documented scientifically. The study can be replicated in other settings with ease. There are a few grammatical errors which need to attended to otherwise this is an excellent piece

6. PLOS authors have the option to publish the peer review history of their article (what does this mean? ). If published, this will include your full peer review and any attached files.

**Do you want your identity to be public for this peer review?** For information about this choice, including consent withdrawal, please see our Privacy Policy .

Reviewer #1: **Yes: ** Dr Chipo Chimamise

---

## [Editor Report · Acceptance letter]

PONE-D-24-59015

PLOS ONE

Dear Dr. Rai,

I'm pleased to inform you that your manuscript has been deemed suitable for publication in PLOS ONE. Congratulations! Your manuscript is now being handed over to our production team.

Kind regards,

on behalf of

Dr. Alfredo Luis Fort

Academic Editor

PLOS ONE